# Dual-objective Language Models: Training Efficiency Without Overfitting

**David Samuel**[*]
University of Oslo
davisamu@uio.no

**Lucas Georges Gabriel Charpentier**[*]
National Library of Norway
lucas.charpentier@nb.no

## Abstract

This paper combines autoregressive and masked-diffusion training objectives without any architectural modifications, resulting in flexible language models that outperform single-objective models. Autoregressive modeling has been a popular approach, partly because of its training efficiency; however, that comes at the cost of sensitivity to overfitting. On the other hand, masked-diffusion models are less efficient to train while being more resilient to overfitting. In this work, we demonstrate that dual-objective training achieves the best of both worlds. To derive the optimal balance between both objectives, we train and evaluate 50 language models under varying levels of data repetition. We show that it is optimal to combine both objectives under all evaluated settings and that the optimal balance is similar whether targeting autoregressive or masked-diffusion downstream performance.

## 1 Introduction

The dominant paradigm for training recent language models has been *autoregressive* next-token prediction (Brown et al., 2020). This approach is remarkably efficient in training, allowing models to quickly absorb vast amounts of text. However, this comes with a significant drawback: a tendency to overfit when training data is repeated (Muennighoff et al., 2023). This issue is becoming increasingly critical as the community reaches the so-called *data wall* – the imminent exhaustion of available data required to train ever-larger models according to established scaling laws (Villalobos et al., 2024).

An alternative approach, *masked-diffusion* language modeling, offers a compelling solution to the overfitting problem (Prabhudesai et al., 2025; Ni et al., 2025). Yet, this robustness comes at the cost – these models are known to be much less sample-efficient than their autoregressive counterparts (Nie et al., 2025a). The complementary strengths of these two approaches suggest combining them as a natural solution to counteracting their failure modes.

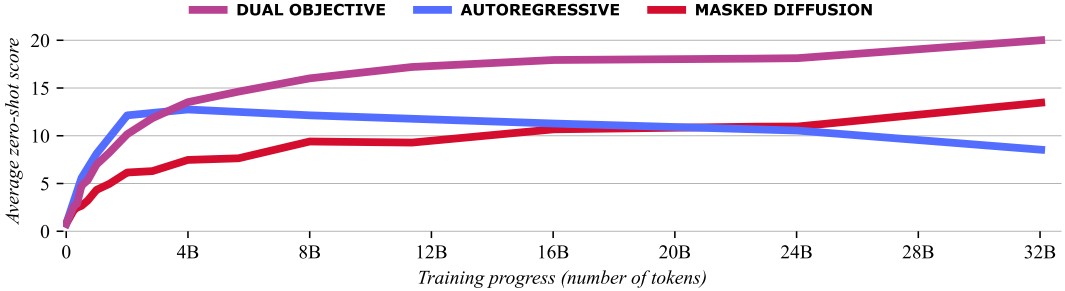

Figure 1: **The dynamics of zero-shot performance.** The three models are trained in a rather extreme setting – 128 repetitions of the training corpus. The autoregressive objective (blue line) converges the fastest but also very quickly overfits; the masked-diffusion objective (red line) converges slowly but without being negatively affected by the high amount of repetitions. Combining both objectives together (purple line) results in fast convergence as well as robustness to overfitting.

---

[*]Equal contribution. Work done at the Language Technology Group, University of Oslo.

In this work, we show that it is possible to achieve the best of both worlds by simultaneously training a single language model on both autoregressive and masked-diffusion objectives. The core idea is to use the training efficiency of the autoregressive objective for rapid initial learning while using the masked-diffusion objective to regularize the model and prevent it from overfitting. The effectiveness of this dual-objective approach is illustrated in Figure 1. In the extreme data-constrained setting with 128 data repetitions, the purely autoregressive model learns quickly but then catastrophically overfits. The masked-diffusion model is immune to overfitting but converges very slowly. Our proposed dual-objective model combines the strengths of both and successfully leverages the given compute and data. The resulting models can be deployed as a standard autoregressive models with no inference overhead.

Building on this observation, we conduct a large-scale systematic study to find the optimal balance between these two objectives under varying degrees of data constraints. Our main contributions are:

- We propose a dual-objective training method that combines autoregressive and masked-diffusion losses, enabling a single model to excel at both unidirectional and bidirectional tasks.

- Through an extensive empirical study, we systematically map the relationship between data repetition, the ratio of training objectives, and final downstream performance. Demonstrating that our dual-objective approach is superior to single-objective training in all evaluated settings, for both autoregressive and masked-diffusion evaluation – including the finding that dual-objective models outperform pure masked-diffusion models even in regular data settings.

- We derive two practical recommendations for setting the optimal objective ratio when training in both regular and data-constrained regimes, providing concrete guidelines for future training of large language models.

## 2 BACKGROUND

Before diving into details of combining autoregressive and masked-diffusion models, we need to briefly describe those two modeling approaches and language modeling in general. As the name suggests, *language models* are statistical models $p_{\boldsymbol{\theta}}(\cdot)$ of the true language distribution of some training corpus $\mathcal{D}$. The training corpus consists of sequences $\boldsymbol{x} = (x_1, x_2, \ldots x_N) \in \mathcal{D}$ of subword tokens. The language models are trained by finding such parameters $\boldsymbol{\theta}$ that maximize the likelihood estimation (MLE; Fisher, 1922; 1925):

$$\operatorname*{argmax}_{\boldsymbol{\theta}} \ \mathbb{E}_{\boldsymbol{x} \sim \mathcal{D}} \left[ \log p_{\boldsymbol{\theta}}(\boldsymbol{x}) \right]. \tag{1}$$

In this paper, we combine two popular approaches for computing $p_{\boldsymbol{\theta}}(\cdot)$, *autoregressive language modeling* and *masked-diffusion language modeling*.

### 2.1 AUTOREGRESSIVE LANGUAGE MODELING

Language models have a long tradition and since their inception in the seminal paper by Shannon (1951), they have been factored into a chain of next-token prediction terms $p_{\boldsymbol{\theta}}(x_i \mid \boldsymbol{x}_{<i})$:

$$-\log p_{\boldsymbol{\theta}}(\boldsymbol{x}) = -\sum_{i=1}^{|\boldsymbol{x}|} \log p_{\boldsymbol{\theta}}(x_i \mid \boldsymbol{x}_{<i}) \stackrel{\text{def}}{=} \mathcal{L}_{\text{AR}}(\boldsymbol{x}; \boldsymbol{\theta}). \tag{2}$$

Computation of the next-token likelihoods can be efficiently parallelized when modeled by transformer networks (Vaswani et al., 2017), and thanks to their scalability, it has been the most popular paradigm behind the recent era of large language models (Brown et al., 2020).

### 2.2 MASKED-DIFFUSION LANGUAGE MODELING

Masked-diffusion language models have recently become a popular alternative to autoregressive models (Austin et al., 2021; Lou et al., 2024; Sahoo et al., 2025; Ou et al., 2025; Nie et al., 2025b). Computing $p_{\boldsymbol{\theta}}(\cdot)$ with masked-diffusion is slightly more complicated than with autoregression, but the resulting language model learns to handle full *bidirectional* context, which can lead to increased performance on downstream tasks (Berglund et al., 2024; Samuel, 2025).

First, following Austin et al. (2021), we define the forward (and backward) diffusion process that gradually turns a sequence of tokens $\boldsymbol{x}$ into special mask tokens (and vice-versa). The diffusion process $\{\boldsymbol{x}^t\}$ depends on the time variable $t \in [0, 1]$ so that $\boldsymbol{x}^{(0)} = \boldsymbol{x}$ and $\boldsymbol{x}^{(1)}$ is a fully masked sequence. The intermediate values are defined by the probability distribution $q$:

$$q_{t|0}(\boldsymbol{x}^t \mid \boldsymbol{x}) \overset{\text{def}}{=} \prod_{i=1}^{|\boldsymbol{x}|} q_{t|0}(x_i^t \mid x_i); \text{ where } q_{t|0}(x_i^t \mid x_i) \overset{\text{def}}{=} \begin{cases} 1 - t, & x_i^t = x_i, \\ t, & x_i^t = \mathsf{mask}. \end{cases} \tag{3}$$

We can see that each token can either remain unchanged or turn into a mask token with probability $t$. The forward process is fully reversible and we can accordingly define the backward process, which gradually unmasks a sequence (Austin et al., 2021). Using the results from Ou et al. (2025), the probability distribution $q_{0|t}(x_i|\boldsymbol{x}^t)$ governing the backward process can be modeled with a time-independent transformer language model with parameters $\boldsymbol{\theta}$ as $p_{\boldsymbol{\theta}}(x_i \mid \boldsymbol{x}^t)$. This model can be fitted to the training data by minimizing the upper bound on the negative log-likelihood estimate (Ou et al., 2025):

$$-\log p_{\boldsymbol{\theta}}(\boldsymbol{x}) \leq -\int_0^1 \mathop{\mathbb{E}}_{\boldsymbol{x}^t \sim q_{t|0}(\cdot|\boldsymbol{x})} \left[ \frac{1}{t} \sum_{\{i|x_i^t=\mathsf{mask}\}} \log p_{\boldsymbol{\theta}}(x_i \mid \boldsymbol{x}^t) \right] \mathrm{d}t \overset{\text{def}}{=} \mathcal{L}_{\text{MD}}(\boldsymbol{x}; \boldsymbol{\theta}). \tag{4}$$

The integral can be equivalently written as the expectation over $t \sim \mathcal{U}(0, 1)$, thus, it can be directly used as a training objective when estimated by Monte-Carlo sampling (Metropolis & Ulam, 1949). Such a Monte-Carlo estimate can also be used at inference-time for likelihood-based evaluation, similarly to Equation (2). Note that the resulting objective is very similar to the one used to train masked language models such as BERT (Devlin et al., 2019).

## 3 DUAL-OBJECTIVE LANGUAGE MODELING

The method of combining autoregressive and masked (diffusion) objectives is mostly based on the earlier GPT-BERT approach by Charpentier & Samuel (2024). They showed promising results for very small language models trained within the limitations of the BabyLM Challenge (Hu et al., 2024). We extend their approach to masked-diffusion language models and to orders of magnitude larger computation scale.

Dual-objective language models are trained by minimizing the following combined loss function, which is further explained below in more detail:

$$\underset{\boldsymbol{\theta}}{\arg\min} \mathop{\mathbb{E}}_{\boldsymbol{x} \sim \mathcal{D}} \Big[ \alpha \mathcal{L}_{\text{AR}}(\boldsymbol{x}; \boldsymbol{\theta}) + (1 - \alpha) \mathcal{L}_{\text{MD}}(\boldsymbol{x}; \boldsymbol{\theta}) \Big]. \tag{5}$$

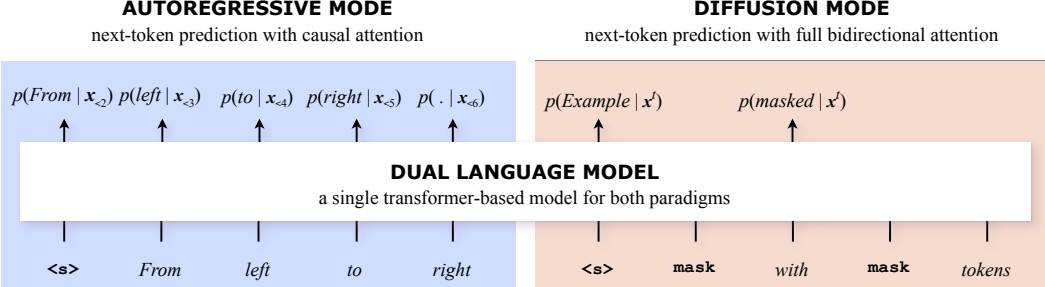

Figure 2: **Two modes of operation inside a single model.** We use the same transformer architecture with the same parameters to do both diffusion and autoregression language modeling, the only difference between the two modes is the input sequence and the attention mask.

**Loss weighting**    The balance between the autoregressive objective $\mathcal{L}_{AR}$ and masked-diffusion objective $\mathcal{L}_{MD}$ is controlled by the hyperparameter $\alpha$. It is crucial for controlling the trade-off between training efficiency and overfitting robustness; its relation to the number of data repetitions is extensively tested by the following experiments.

In practice, naively mixing both objectives within a single batch could result in reduced throughput. For this reason, we assign each GPU device to a single objective so that the computation graph remains simple and static, and can be efficiently compiled. To be specific, we distribute the training of each model across 256 devices, which allows for choosing between $256 + 1$ values: $\alpha \in \{i/256 \,|\, i = 0, 1, \ldots 256\}$.

**Diffusion as next-token prediction**    Our goal is to align $\mathcal{L}_{AR}$ and $\mathcal{L}_{MD}$ so that they can be parameterized by a single transformer model. For this reason, we use a slightly modified version of masked language modeling called *masked next-token prediction* (MNTP; Lv et al., 2024). With this approach, the model always uses the hidden state at position $i$ to predict the next token at position $i + 1$ (we prove that this parameterization is as expressive as the standard approach in Appendix F). In this way, both modes of operation are unified as they both perform next-token prediction, as illustrated in Figure 2. MNTP has also been used in recent work for adapting a masked diffusion model from an autoregressive checkpoint (Gong et al., 2025; Ye et al., 2025).

**Standard transformer architecture**    The main benefits of using masked next-token prediction are that we can use exactly the same transformer architecture as standard autoregressive models, and we can optimize its parameters with both objectives at the same time. The only difference between the two modes of operation is the inputs – they are either (partially) masked inputs with empty (fully bidirectional) attention masks, or full unchanged inputs with causal (unidirectional) attention masks.

## 4    EVALUATION

While it is common practice to only consider the value of loss on a held-out set when evaluating language models (Kaplan et al., 2020; Hoffmann et al., 2022; Muennighoff et al., 2023), it is important to measure the actual downstream performance to accurately assess the effect of different training configurations. This is especially crucial when training with two incompatible training losses. That being said, we also report validation losses in Appendix G.

**Tasks**    We evaluate our models on nine standard language modeling tasks in a zero-shot fashion. All tasks consist of a context (which can be empty) and multiple different completions where one is correct and the others are incorrect. We evaluate the sum of the log-likelihood of each completion and assign the completion with the maximum sum as the prediction of the model. Table 1 lists the tasks:

Table 1: **The list of evaluation tasks.** The ARC[†] datasets contain some examples with 3 or 5 completions rather than 4. All tasks are evaluated zero-shot.

| Task | # Examples | # Completions | Split | Reference |
| --- | ---: | ---: | --- | ---: |
| ARC-Easy (ARC-E) | 2 376 | 4[†] | test | Clark et al. (2018) |
| ARC-Challenge (ARC-C) | 1 172 | 4[†] | test | Clark et al. (2018) |
| BLiMP | 67 000 | 2 | — | Warstadt et al. (2020) |
| Commonsense QA (CSQA) | 1 221 | 5 | val | Talmor et al. (2019) |
| HellaSwag (HSwag) | 10 042 | 4 | val | Zellers et al. (2019) |
| MMLU | 14 042 | 4 | test | Hendrycks et al. (2021) |
| OpenBook QA (OBQA) | 500 | 4 | test | Mihaylov et al. (2018) |
| Physical Interaction QA (PIQA) | 1 838 | 2 | val | Bisk et al. (2020) |
| Social IQa (SIQA) | 1 954 | 3 | val | Sap et al. (2019) |

**Evaluation setup**    We follow the guidelines of the OLMES paper (Gu et al., 2025) for the normalization of our log-likelihood estimations as well as the prompt format, with two changes: 1) we only evaluate in a zero-shot fashion to simplify the setup, 2) we only consider their "cloze" formulation of each task, which is more suitable for smaller models. For the BLiMP task, which is not considered in the OLMES evaluation suite, we do not apply any length normalization and score each sample with

the raw log-likelihood score. Since the BLiMP and MMLU tasks contain multiple sub-tasks (67 for BLiMP, and 57 for MMLU), we report their macro-average as the final score. More information on how each task is normalized can be found in Appendix C.

**Normalized score averaging** To ensure a fair aggregation of the different task scores, we first normalize the scores such that the random baseline of each task is at $0$ and the maximum is at $1$; similarly to the Open LLM Leaderboard (Fourrier et al., 2024). To achieve this we apply the following formula to our scores: $\text{score}(x, t) = (x - r_t)/(m_t - r_t)$, where $x$ is the raw score, $r_t$ is the random baseline and $m_t$ is the optimal score for task $t$. We then take the simple average of the normalized scores across all tasks as the final performance of our model.

## 4.1 AUTOREGRESSIVE (UNIDIRECTIONAL) EVALUATION

To evaluate the autoregressive capabilities of our models, we use Equation (2) to estimate the log-likelihood of each completion. Specifically, given a completion ($\boldsymbol{w}$) and context ($\boldsymbol{c}$), we calculate the conditional log-likelihood as $\log p_{\boldsymbol{\theta}}(\boldsymbol{w} \mid \boldsymbol{c}) = \sum_i \log p_{\boldsymbol{\theta}}(w_i \mid \boldsymbol{c}, \boldsymbol{w}_{<i})$.

## 4.2 MASKED-DIFFUSION (BIDIRECTIONAL) EVALUATION

One possibility to evaluate the masked-diffusion capabilities of our models is to also leverage the training objective in Equation (4) and estimate the conditional log-likelihood of each completion by Monte-Carlo sampling. We describe this approach in more detail in Appendix D. While it provides accurate downstream scores, it is computationally expensive and less accurate than using simpler pseudo log-likelihood (PLL; Wang & Cho, 2019; Salazar et al., 2020; Samuel, 2025) estimation.

PLL allows us to do bidirectional evaluation more than ten times faster while being more accurate than Monte-Carlo sampling (Appendix J). Therefore, we use PLL for evaluating the bidirectional capability of our models. We fully describe this method in Appendix E. As visualized in Figure 3 on the left, we specifically use the semi-autoregressive variation of PLL proposed by Samuel (2025).

Figure 3: **Visual representations of bidirectional evaluation methods.** Pseudo log-likelihood estimation (on the left) reaches accurate likelihood scores substantially faster than the (theoretically grounded) Monte-Carlo estimation (on the right).

# 5 EXPERIMENTS

## 5.1 PRETRAINING SETUP

We train each 470-million-parameter language model (with 360M non-embedding weights) on 32 billion tokens in total. A repetition factor $R$ means we sample a unique subset of size $^{32\text{B}}/_R$ tokens and repeat it $R$ times during training. This total token budget is more than $4\times$ past the Chinchilla compute-optimal point (Hoffmann et al., 2022); we specifically decided to conduct the experiments in this regime as it reflects how modern language models are trained in practice. This compute budget is also large enough to induce non-trivial zero-shot downstream performance, enabling us to measure clear differences between different configurations.

**Model architecture** The language models have 24 layers with hidden size of $1\,024$, their self-attention operations are divided into 16 parallel heads, the feed-forward modules have intermediate size of $3\,554$, and the vocabulary is set to $51\,200$ tokens. As for the architecture itself, we follow the usual modifications of the original transformer recipe (Vaswani et al., 2017) – pre-normalization

(Nguyen & Salazar, 2019) with RMSNorm (Zhang & Sennrich, 2019), rotational positional embedding (Su et al., 2024) and Swish-gated linear units (Ramachandran et al., 2018; Shazeer, 2020).

**Optimization**    The parameters are optimized by the Muon optimizer for faster convergence (Jordan et al., 2024), specifically its variation proposed by Liu et al. (2025). The learning rate is set to 0.007 and decayed according to the warmup-stable-decay (WSD; Hägele et al., 2024) schedule (without warmup steps and 2 048 steps of linear decay). In total, each model is trained for 8 192 steps with 4M tokens in each global batch and with a sequence length of 2 048 tokens. The optimization is regularized by weight decay (with strength of $10^{-1}$) and by an auxiliary z-loss term (with strength of $10^{-4}$; Chowdhery et al., 2022).

**Training corpus and tokenizer**    Even though we limit the training data to 32B tokens, we deliberately choose a text corpus that is not excessively filtered and that is representative of large-scale web crawls used in practice. We randomly sample English documents with 32B tokens in total from the HPLT v2 corpus (Burchell et al., 2025), which combines extracted webpages from the Internet Archive and CommonCrawl. We also use a smaller disjoint subset to monitor the validation loss. To prevent a potential bias from using an external tokenizer, we train a standard byte-level BPE tokenizer (Gage, 1994) with 51 200 subwords directly on the full training data.

## 5.2    SEARCHING FOR THE OPTIMAL $\alpha$

We trained and evaluated 50 language models in total, the results are plotted in Figure 4. In order to deal with the noisy nature of this data and to better understand the relation between the amount of data repetitions and the optimal $\alpha$, we use simple statistical models.

**Interpolation with Gaussian process**    Specifically, we use Gaussian process regression (GPR; Williams & Rasmussen, 1995) with a composite kernel structure to model the relationship between data repetitions, $\alpha$ and downstream performance. The composite kernel consists of a constant kernel multiplied by an anisotropic Matérn kernel ($\nu = 1.5$; Stein, 1999) combined additively with a white noise kernel to account for observation noise. The input features are standardized to zero mean and unit variance, and the output features are normalized. The kernel parameters are optimized by L-BFGS-B (Liu & Nocedal, 1989) using SciPy (Virtanen et al., 2020). The resulting interpolations in Figure 4 show regular structure while closely fitting the data with $R^2$ over 0.99 in all cases.

**The optimal autoregressive-diffusion ratios**    The fitted Gaussian process is a probabilistic model of the downstream performance with regard to the amount of data repetition and $\alpha$. Thus, we can transform this to the probability that a particular $\alpha$ is optimal for the given data repetition. More concretely, we can estimate the density of this distribution by sampling from the posterior of the GPR model. The result of this is visualized in the bottom part of Figure 4.

## 5.3    RESULTS AND DISCUSSION

The structure of Figure 4 becomes clearer once we identify which training settings result in overfitting during training.[1] The density of optimal $\alpha$ weights highlights that there are two regions to consider: *1) Regular-data region* where a language model trained solely on the autoregressive objective does not overfit – this roughly corresponds to 16 repetitions of training data and less, as also shown by Muennighoff et al. (2023). *2) Data-constrained region* – roughly corresponding to 32 data repetitions and more – where overfitting is an important consideration.

In the first case, it is clearly beneficial to put more weight on the autoregressive training than on masked-diffusion. Yet, training only autoregressively does not lead to any improvement in any experiments within the regular-data region. Even when evaluated purely autoregressively, the differences between $\alpha$ set to 1 and $^{15}/_{16}$ are negligible. Switching to bidirectional evaluation, the single-objective $\alpha = 1$ performs poorly while all models trained with $\alpha$ values between $^{255}/_{256}$ and $^{15}/_{16}$ perform similarly – notably, they all substantially outperform models trained only with masked-diffusion. This is a key finding: even without any data constraint, the dual-objective models achieve stronger masked-diffusion performance than pure masked-diffusion training, despite dedicating only a small fraction of training to the masked-diffusion objective. We hypothesize that the prevalence of

---

[1]Here, *overfitted training runs* are those runs, in which the held-out loss starts diverging while the training loss keeps converging (Appendix G). Such runs are highlighted in Figure 4 by $\times$ marks.

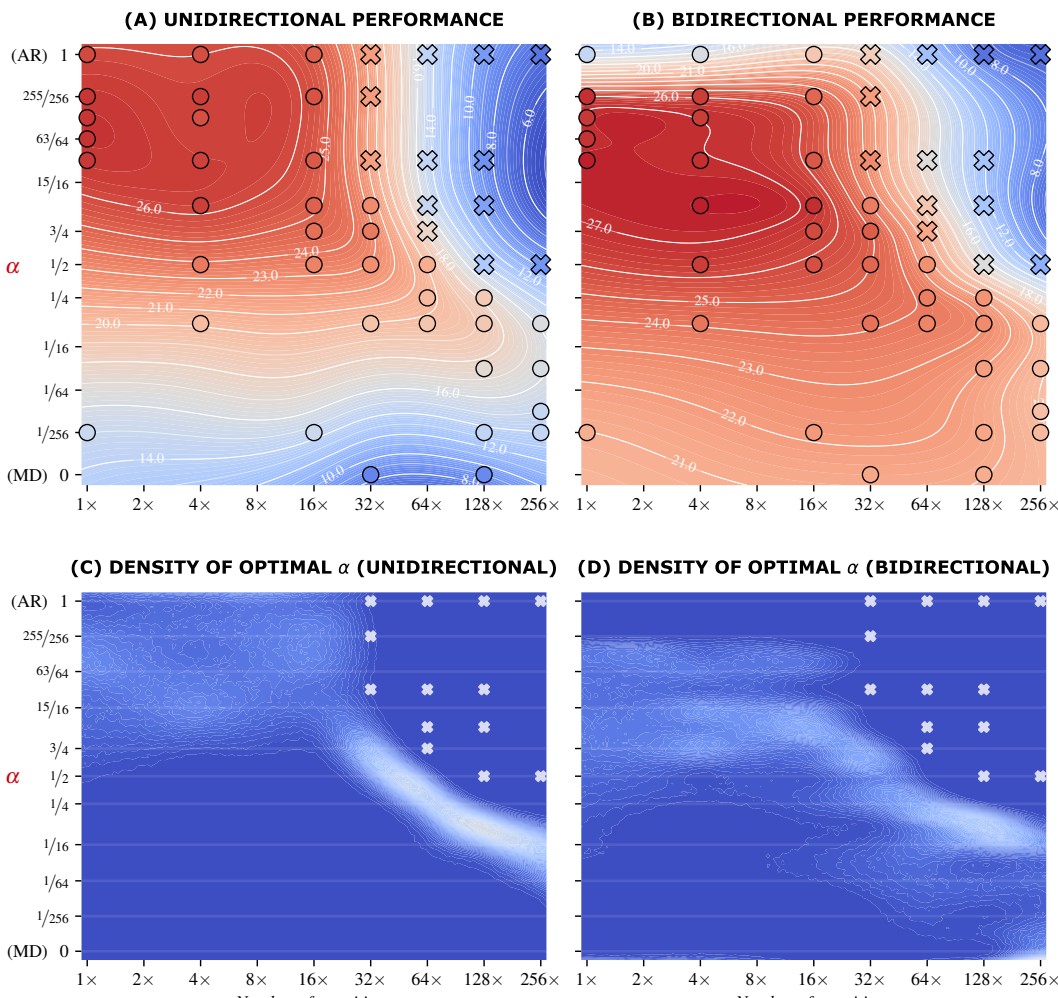

Figure 4: **Interpolated unidirectional and bidirectional results.** The (a) and (b) figures on top show the relation between repetitions (x-axis) and the autoregressive-diffusion weight $\alpha$ (y-axis); the contours follow the Gaussian process model that interpolates the average performance of language models trained according to the specified settings. The respective results are plotted either as crosses when the model overfitted during training, or as circles. The (c) and (d) figures below visualize the estimated probability that a particular $\alpha$ (y-axis) is optimal for a given number of repetitions (x-axis).

the autoregressive objective leads to fast convergence, and that the small amount of masked-diffusion balances its slower convergence by inducing useful modeling priors. This leads us to formulating the first practical recommendation:

**Remark 1** *Language modeling under regular data settings*

When training a language model in a regular data setting (16 repetitions or less), train with a small amount of masked-diffusion objective ($\alpha \approx {}^{63}/_{64}$) to achieve stronger bidirectional performance than pure masked-diffusion training without losing any autoregressive performance.

In the second data-constrained case, the relation between data repetition, $\alpha$, and final performance seems more complicated. We risk overfitting by putting too much weight on autoregression and underfitting by focusing too much on masked-diffusion; as evident from Figure 4, the interval of optimal $\alpha$ values is fairly narrow. On the other hand, the optimal values are surprisingly similar for the unidirectional and bidirectional performance. We can notice that the region of optimal $\alpha$ values

is right beneath the region of $\alpha$ values that lead to overfitting, but the question is how to identify such an $\alpha$. It is possible to have an alternative interpretation of the autoregressive-diffusion weights and count the number of data repetitions that each objective is individually trained on – then we can see that more than 32 autoregressive repetitions lead to overfitting while fewer than 8 autoregressive repetitions lead to underfitting. Thus, based on the empirical results, our recommendation for this scenario is:

---

**Remark 2**  *Data-constrained language modeling*

When training a language model in a data-constrained setting (more than 32 repetitions), choose $\alpha$ that exposes the autoregressive objective to roughly 16 repetitions of the training data.

---

Table 2: **The normalized autoregressive performance of selected models.** We show the results on all nine evaluated tasks for three repetition values; each repetition group contains the results of the best-performing $\alpha$ and of the autoregressive-only model. The scores for each task are normalized so that 0% corresponds to random baseline and 100% is the perfect score. The best result for each dataset size is boldfaced.

| Model configuration | ARC-C | ARC-E | BLiMP | CSQA | HSwag | MMLU | OBQA | PIQA | SIQA | Average |
|---|---|---|---|---|---|---|---|---|---|---|
| 1 REPETITION | | | | | | | | | | |
| Dual ($\alpha = {}^{63}/_{64}$) | 5.7 | 28.6 | **63.7** | **35.1** | 31.1 | **4.9** | **17.6** | **40.9** | 14.3 | **26.9** |
| Autoregressive ($\alpha = 1$) | **5.9** | **30.3** | 61.3 | 33.5 | **31.7** | 3.8 | 13.6 | 39.4 | **15.2** | 26.1 |
| 32 REPETITIONS | | | | | | | | | | |
| Dual ($\alpha = {}^{3}/_{4}$) | 3.3 | **28.0** | **57.9** | **31.1** | **26.4** | 3.6 | **14.4** | **36.1** | **14.6** | **23.9** |
| Autoregressive ($\alpha = 1$) | **5.0** | 24.9 | 53.3 | 28.5 | 25.4 | **3.8** | 9.9 | 33.3 | 14.2 | 22.0 |
| 128 REPETITIONS | | | | | | | | | | |
| Dual ($\alpha = {}^{1}/_{8}$) | **1.7** | **23.6** | **56.1** | **24.8** | **14.2** | **1.6** | **8.5** | **28.1** | **13.3** | **19.1** |
| Autoregressive ($\alpha = 1$) | -1.0 | 12.3 | 33.2 | 6.8 | 8.1 | 1.1 | -0.5 | 15.8 | 8.9 | 9.4 |

**Generalization to larger language models**  An obvious question is whether the recommendations hold even at much bigger scale for larger language models. Reliably answering this question would require expensive experimentation, but we believe that the conclusions hold for two reasons. Firstly, according to our results, the optimal $\alpha$ values are clearly correlated with overfitting of autoregressive language models. Since the overfitting behavior does not depend on model size according to previous work (Muennighoff et al., 2023; Prabhudesai et al., 2025), we believe that the optimal $\alpha$ values should also not change. Secondly, the relative burden of representing two modes of operation within the learned parameters decreases with model size, so we believe that the benefit of the dual training objective should actually increase with model size.

**Detailed results**  To put the abstract average scores into another perspective, we look at the individual (normalized) scores per task in Table 2. The results show that the improvement in performance from using a dual objective is observed on a majority of tasks. This is especially true the more repetitions there are. The detailed scores also highlight how effectively the dual objective learns from limited data, reaching nontrivial performance even when exposed to just 256M tokens of training data (under 128 repetitions). We observe similar trends for masked-diffusion evaluation except that as the number of repetitions decreases, the performance gap increases rather than decreases. Detailed performance for the masked-diffusion evaluation can be found in Appendix L.

## 5.4  GENERALIZATION TO PREFIX LANGUAGE MODELING

Prefix language modeling (Dong et al., 2019; Raffel et al., 2020; Wang et al., 2022) is a promising alternative to the two training objectives investigated in this work. It processes the conditioning part (prefix, $c$ in notation from Section 4.1) of a text fully bidirectionally while the completion part ($w$ in Section 4.1) is processed autoregressively. Given that our models are trained with both unidirectional

and bidirectional attention, we test whether the exposure to both can induce generalization to prefix language modeling without any further training. We repeat the earlier autoregressive evaluation with prefix attention masks and plot the results in Figure 5.

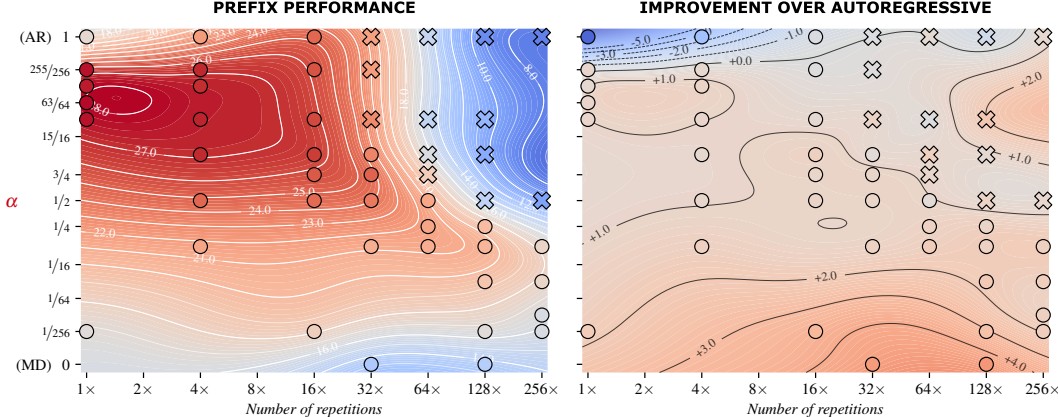

Figure 5: **Interpolated prefix results.** The figures show the relation between data repetitions (x-axis), $\alpha$ (y-axis), and downstream performance (color-coded). The individual results are interpolated by a GPR model. The right figure demonstrates the relative improvement of prefix-masked evaluation compared to fully unidirectional evaluation (blue color denotes decreased performance and red color denotes a performance increase).

The right side of Figure 5 shows the overall improvement of the prefix evaluation over the autoregressive one. Notably, we can see that it is reliably over one percentage point better across most configurations that combine both training objectives. This finding leads to our third recommendation:

> **Remark 3**  *Induced prefix language modeling*
>
> The autoregressive performance of dual-objective language models can be reliably improved at inference time by processing the conditional part of a prompt fully bidirectionally.

## 6 RELATED WORK

**Combining autoregressive and masked (diffusion) language modeling**    This paper builds upon the GPT-BERT training objective by Charpentier & Samuel (2024), validating its effectiveness in a more practical setting. However, there is a long history of papers that tried to combine bidirectional masked language modeling with unidirectional autoregressive modeling: T5 (Raffel et al., 2020) and BART (Lewis et al., 2020) were the first to train with autoregressive fill-in-the-blank training objectives by relying on encoder-decoder transformer architectures. Later, Du et al. (2022) proposed GLM, which uses the same objective as T5 while using a simpler decoder-only architecture with a complicated scheme of positional encodings. CM3 by Aghajanyan et al. (2022) further simplifies training by not requiring any non-standard architectural modifications like the previous work. As they also add autoregressive language-modeling objective, their work is close to our approach – a model trained with CM3 can be used as any other autoregressive model at inference time, similarly to us. However, our objective also generalizes masked-diffusion language modeling and allows for fine-grained balance of the two objectives throughout training. More recently AntLM by Yu et al. (2024) proposed to switch from one objective to the other in a curriculum fashion, starting with a short autoregressive training, followed by a long masked language training and finishing on another short autoregressive training. While this does show promise, the transition from one objective to the other leads to forgetting of the previous objective whereas our objective continuously learns both objectives. Other notable works include prefix language models (Dong et al., 2019; Raffel et al., 2020; Wang et al., 2022) and UL2 (Tay et al., 2023).

**Scaling of autoregressive and masked-diffusion models**   Concurrent works by Prabhudesai et al. (2025) and Ni et al. (2025) have demonstrated that masked-diffusion models outperform autoregressive models in data-constrained training regimes. Our results confirm their findings but we show that using either of these training objectives is never optimal – combining them together should always be better, not only in data-constrained settings.

**Bidirectional masking of user and system prompts**   A recent paper by Katz et al. (2025) shows that using a bidirectional mask on user and system prompts improves performance on a wide variety of tasks, in line with Section 5.4. However, for models to be able to use such masks, the authors first need to train adapters. Our work shows that by training both autoregressive and masked-diffusion at the same time, we are able to induce the prefix mask without any additional training.

**Data-constrained scaling laws**   Muennighoff et al. (2023) studies the scaling laws of autoregressive models in data-constrained settings with a similar motivation to this paper. They show that autoregressive models cannot meaningfully learn from more than 16 data repetitions – we demonstrate that this value is at least an order of magnitude larger when training with the dual objective.

**Autoregressive diffusion**   Our work shares motivation with the autoregressive-diffusion models proposed by Wu et al. (2023). The diffusion process in that work is biased towards left-to-right denoising, which improved the decoding efficiency of the diffusion language models at that time. Similarly, Arriola et al. (2025) speeds up decoding of masked-diffusion models by autoregressively generating chunks of tokens where each chunk is decoded by a diffusion process. In both cases, the resulting models are still diffusion models – albeit faster; these approaches do not generalize over autoregressive and masked-diffusion language modeling as our method.

**Fair MD-AR comparison**   The recent work by Xue et al. (2025) modifies masked-diffusion language models by parameterizing them with causally-masked transformers, which makes the diffusion models more comparable to standard autoregressive models – decoupling their architectural differences from differences in training objectives. Their conclusion is that masked diffusion alone is a suboptimal objective for language, which is also confirmed by our experiments (Figure 4). However, we found that by simply combining both objectives, we can get the benefits of diffusion without losing any performance.

**Approaching the data wall**   Large language models are known to reliably follow the empirical *scaling laws* that describe how their performance should improve with increased compute, model size, and training data. Kaplan et al. (2020) first demonstrated these relationships, showing how the training loss decreases as a power law with respect to these three parameters. These laws were later refined by Hoffmann et al. (2022), who showed that compute-optimal training requires scaling data and model size together. Related to our work, the scaling laws reveal a fundamental problem: achieving each incremental gain in performance requires exponentially more training data. Thus, data-constrained language modeling is quickly becoming a relevant field of study even for high-resource languages such as English.

## 7   CONCLUSION

In this work, we addressed the fundamental trade-off between the training efficiency of autoregressive models and the overfitting resilience of masked-diffusion models. We have empirically demonstrated that a dual-objective training strategy successfully achieves the best of both worlds, resulting in models that converge rapidly without any performance degradation in data-constrained settings. Crucially, because this unification requires no architectural changes, the resulting models incur no inference overhead and can be deployed as standard autoregressive transformers.

We established that combining objectives is universally beneficial and derived practical guidelines for selecting the optimal $\alpha$ based on the degree of data repetition. Furthermore, we observed that the diffusion objective induces robust prefix language modeling capabilities, leading to superior performance on downstream tasks compared to standard autoregressive baselines. While training on hundreds of data repetitions may seem extreme today, the asymmetry between exponentially scaling compute budgets and the finite supply of high-quality text suggests that data constraints will become increasingly relevant for frontier model development. Our findings indicate that dual-objective training provides a robust and compute-efficient path forward that retains standard inference capabilities as the field approaches these fundamental limits.

## REPRODUCIBILITY STATEMENT

To ensure reproducibility of our work we provided the guidelines on how to train language models on both objectives at the same time in Section 3. For our model parameters and hyperparameters we specified those in Section 5.1. We describe how we perform the evaluations, the number of mask tokens used for PLL, the prompt formats, and log-likelihood normalizations in Section 4, Appendix C, and Appendix E. We openly release our custom training and evaluation code at `https://github.com/ltgoslo/dual-language-models`. The training code is based on the common and freely distributed deep-learning framework PyTorch (Paszke et al., 2019). The trained models are released openly under the Apache 2.0 license for further investigation at `https://huggingface.co/ltg/dual-lm-470m`.

## AUTHOR CONTRIBUTIONS

Both authors have contributed equally and should be considered shared first authors of this manuscript.

## ACKNOWLEDGMENTS

The computations were performed on resources provided through Sigma2 – the national research infrastructure provider for high-performance computing and large-scale data storage in Norway. We acknowledge Norway and Sigma2 for awarding this project access to the LUMI supercomputer, owned by the EuroHPC Joint Undertaking, hosted by CSC (Finland) and the LUMI consortium through project 465001890.

The efforts described in this paper were jointly funded by the University of Oslo and the HPLT project (High Performance Language Technologies; coordinated by Charles University).

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

## A    THE USE OF LARGE LANGUAGE MODELS

Large language models have been used to provide feedback, fix grammatical errors and improve the writing in this paper; in particular, we used the Claude family of language models from `https://claude.ai`. In addition, we used the autocompletion tool from GitHub Copilot when writing the code used in this work.

## B    ERRATUM: LOSS FORMULATION

The original version of this paper as well as the training code used the following formulation the loss function:

$$\operatorname*{argmin}_{\boldsymbol{\theta}} \ \mathbb{E}_{\boldsymbol{x} \sim \mathcal{D}} \Big[ 2\alpha \mathcal{L}_{\mathrm{AR}}(\boldsymbol{x}; \boldsymbol{\theta}) + (1 - \alpha)\mathcal{L}_{\mathrm{MD}}(\boldsymbol{x}; \boldsymbol{\theta}) \Big]. \tag{6}$$

The additional factor of two for $\mathcal{L}_{\mathrm{AR}}$ was supposed to balance the $1/t$ term applied when computing $\mathcal{L}_{\mathrm{MD}}$ (recall Equation (4)). This is however not mathematically correct (Equation (4)) and unnecessary; the loss function is thus simplified in this updated version.

## C    LOG-LIKELIHOOD NORMALIZATION

For the BLiMP task, which is not considered in the OLMES evaluation suite, we do not apply any normalization and take the raw log-likelihood. We also stick to the no-context form of this task, where the whole sentence is considered the completion. We apply character length normalization to ARC-Easy, HellaSwag, MMLU, PIQA, and SIQA. Finally, we apply point-wise mutual information normalization (Holtzman et al., 2021), where the log-likelihood of the context-informed completion is divided by the log-likelihood of the uncontrained context completion, this can be seen in Equation (7), to ARC-Challenge, Commonsense QA, and OpenBook QA.

$$\mathrm{PMI}(\boldsymbol{w}) = \sum_{i=1}^{|\boldsymbol{w}|} \log \left( \frac{p_{\boldsymbol{\theta}}\left(w_i \mid \boldsymbol{c} \oplus \boldsymbol{w}_{<i}\right)}{p_{\boldsymbol{\theta}}\left(w_i \mid \boldsymbol{u} \oplus \boldsymbol{w}_{<i}\right)} \right), \tag{7}$$

where $\boldsymbol{w}$ is the completion, $\boldsymbol{c}$ is the context, and $\boldsymbol{u}$ is the unconstrained context (in our case, this would be "Answer:")

## D    MONTE CARLO ESTIMATION OF LOG-LIKELIHOOD

To evaluate the masked-diffusion capabilities of our models, we use Equation (4) with the same modification as for the autoregressive evaluation as well as an adaptation of Monte-Carlo sampling to estimate the log-likelihood of each completion. Specifically, we estimate the following expected value:

$$\int_0^1 \mathbb{E}_{\boldsymbol{x}^t \sim q_{t|0}(\cdot|\boldsymbol{x})} \left[ \frac{1}{t} \sum_{\{i|x_i^t = \mathrm{mask}\}} \log p_{\boldsymbol{\theta}}(x_i \mid \boldsymbol{x}^t) \right] \mathrm{d}t = \mathbb{E}_{\substack{t \sim \mathcal{U}(0,1) \\ \boldsymbol{x}^t \sim q_{t|0}(\cdot|\boldsymbol{x})}} \left[ \frac{1}{t} \sum_{\{i|x_i^t = \mathrm{mask}\}} \log p_{\boldsymbol{\theta}}(x_i \mid \boldsymbol{x}^t) \right] \tag{8}$$

To reduce the variance of the estimation and get faster convergence, we take the expectation between $N$ equally spaced points between 0 and 1. instead of taking the expectation over $t \sim \mathcal{U}(0,1)$. Yet, accurate estimation still requires $N \geq 256$, which is unbearably slow – especially when compared to simple autoregressive calculation of log-likelihood that requires only a single forward pass.

## E    PSEUDO LOG-LIKELIHOOD ESTIMATION

The base PLL equation can be described by a slight modification of Equation (2):

$$\log p_{\boldsymbol{\theta}}(\boldsymbol{w}) \approx \sum_{i=1}^{|w|} \log p_{\boldsymbol{\theta}}\big(w_i \mid \boldsymbol{c} \oplus w_0 \oplus \cdots \oplus w_{i-1} \\ \oplus \mathrm{mask} \\ \oplus w_{i+1} \oplus \cdots \oplus w_{|\boldsymbol{w}|}\big) \tag{9}$$

This means that instead of doing a single forward pass, we need to do $|\boldsymbol{w}|$ forward passes to estimate the PLL. However, using a single mask token could lead to underestimating the log-likelihood of words split into multiple tokens. Therefore, we can further modify Equation (9) to have a variable (but constant) number of mask tokens after the token we are trying to estimate:

$$
\begin{aligned}
\log p_{\boldsymbol{\theta}}(\boldsymbol{w}) \approx \sum_{i=1}^{|\boldsymbol{w}|} \log p_{\boldsymbol{\theta}}\big(w_i \mid \; & \boldsymbol{c} \oplus w_0 \oplus \cdots \oplus w_{i-1} \\
& \oplus \, \mathsf{mask} \oplus \cdots \oplus \mathsf{mask} \\
& \oplus \; w_{i+n} \oplus \cdots \oplus w_{|\boldsymbol{w}|}\big),
\end{aligned}
\tag{10}
$$

where $n$ represents the number of [MASK] tokens. In our case, we take a combination of two different numbers of mask tokens (1 and 6), by taking the best score of the two for each task. The two values were chosen experimentally, more details on the results of each number of mask tokens can be found in Appendix H.

## F  PROOF OF LEFT-SHIFT CLOSURE

This section proves that when we parameterize masked-diffusion language models as bidirectional transformers with shifted output, we do not lose any expressivity compared to standard non-shifted bidirectional models. We prove it constructively by defining a shift operation in the RASP language (which can then be compiled into an equivalent transformer model).

**Definition 1** (RASP programs)**.** The Restricted Access Sequence Processing language (RASP; Weiss et al., 2021) is a sequence processing language that uses two types of variables: *sequence operators* and *selectors*; and two types of operators: *element-wise* and *select-aggregate* operators. Valid *programs* in RASP are operations on sequence operators formed by a finite composition of element-wise and select-aggregate operators.

- *Sequence operators* represent sequences of values (akin to hidden states in transformer models). `tokens` and `indices` are two pre-defined sequence operators; the first directly returns a sequence of the input tokens ($\mathtt{tokens}(\text{"hello"}) = [\mathtt{h, e, l, l, o}]$, and the second returns the positional indices ($\mathtt{indices}(\text{"hello"}) = [0, 1, 2, 3, 4]$).

- *Selectors* are binary matrices (akin to attention matrices in transformers).

- *Element-wise operators* are arbitrary element-wise transformations on sequence operators (akin to feed-forward layers in transformers). For example $(\mathtt{indices} + 2)(\text{"hello"}) = [2, 3, 4, 5, 6]$.

- *Select-aggregate operators* consist of two sequentially applied operators `select` and `aggregate` (corresponding to the attention operation).

- $\mathtt{select}(\boldsymbol{x}, \boldsymbol{y}, p)$ is an operator defined on two sequence operators $\boldsymbol{x}$ and $\boldsymbol{y}$, and an element-wise boolean operator $p$ defined on two sequence operators; the result is a selector matrix $\boldsymbol{M}$, where $M_{ij} = p(x_i, y_j)$. For example, $\mathtt{select}([0, 1, 2], [1, 2, 3], <)$ results in a upper-triangular $3 \times 3$ binary matrix (selector).

- $\mathtt{aggregate}(\boldsymbol{M}, \boldsymbol{x}; c)$ is an operator defined on a selector $\boldsymbol{M}$, a sequence operator $\boldsymbol{x}$ and a default value $c$ (usually set to $0$ and omitted for convenience). It produces a sequence operator $\boldsymbol{y}$ such that:
$$
y_i = \begin{cases} \frac{1}{|\{j: M_{ij}=1\}|} \sum_{j: M_{ij}=1} x_j, & \text{if } |\{j: M_{ij}=1\}| > 0, \\ c, & \text{otherwise.} \end{cases}
$$

**Fact 1** (RASP-transformer reduction)**.** *For every valid program written in RASP, there exists an equivalent fully-bidirectional transformer model that computes the same per-position operation; see Weiss et al. (2021); Lindner et al. (2023).*

**Definition 2** ($\Sigma$-realizable functions)**.** We consider programs defined on an input alphabet $\Sigma$ with a special token `` $\in \Sigma$. A valid input sequence $\boldsymbol{x} = (x_1, x_2 \ldots x_n) \in \mathcal{X}$ is every sequence where $x_1 = \text{}$ and all $x_i \in \Sigma$. The output space $\mathcal{Y}$ is made of sequences $\boldsymbol{y} = (y_1, y_2 \ldots y_n) \in \mathcal{Y}$,

where every element is a probability distribution over the alphabet $\Sigma$: that is all $y_i \in [0,1]^{|\Sigma|}$ and $\sum_j (y_i)_j = 1$.

A function $f : \mathcal{X} \to \mathcal{Y}$ is *$\Sigma$-realizable* if there exists a transformer whose output on every input $\boldsymbol{x} \in \mathcal{X}$ equals $f(\boldsymbol{x})$ position-wise. Let $\mathcal{R}_\Sigma$ be the class of all $\Sigma$-realizable functions.

**Theorem 1** (Left-shift closure). *$\mathcal{R}_\Sigma$ is closed under unit left-shifts: for every $f \in \mathcal{R}_\Sigma$, there exists $g \in \mathcal{R}_\Sigma$ such that for all $\boldsymbol{x} \in \mathcal{X}$ and $i \in [1, n-1]$: $g(\boldsymbol{x})_i = f(\boldsymbol{x})_{i+1}$ (note that $f(\boldsymbol{x})_1$ and $g(\boldsymbol{x})_n$ are not constrained).*

*Proof.* The proof constructs a suitable function $g \in \mathcal{R}_\Sigma$ for any $f \in \mathcal{R}_\Sigma$. The new function $g$ will mirror function $f$ and then shift its output so that $g(\boldsymbol{x})_i = f(\boldsymbol{x})_{i+1}$, the shift will be constructed in RASP so that $g$ is $\Sigma$-realizable.

Let $f \in \mathcal{R}_\Sigma$ be any $\Sigma$-realizable function and set $T_f$ as a fully-bidirectional transformer that realizes $f$, so $T_f(\boldsymbol{x})_i = f(\boldsymbol{x})_i$ for all valid inputs $\boldsymbol{x} \in \mathcal{X}$ and all positions $i \in [1, n]$.

First, we define a RASP selector $\boldsymbol{S} = \texttt{select(indices} + 1, \texttt{ indices}, =)$, whose entries therefore satisfy $S_{ij} = 1$ iff $j = i + 1$ (each row $i$ selects exactly the next position $i + 1$, and the last row selects none).

Then, for any sequence operator $\boldsymbol{z}$ (possibly vector-valued), we define a RASP program $\texttt{shift}(\boldsymbol{z}) = \texttt{aggregate}(\boldsymbol{S}, \boldsymbol{z}; c)$, where $c$ is arbitrary and can be simply set to $z_n$. By construction of $\boldsymbol{S}$ and the definition of $\texttt{aggregate}$, we have $\texttt{shift}(\boldsymbol{z})_n = c = z_n$ and for every $i \in [1, n-1]$:

$$\texttt{shift}(\boldsymbol{z})_i = \frac{1}{|\{j : S_{ij} = 1\}|} \sum_{j: S_{ij}=1} z_j = z_{i+1}. \tag{11}$$

Using Fact 1, there exists a transformer $T_{\texttt{shift}}$ that computes the RASP program $\texttt{shift}$. Therefore, we can construct a transformer $T_g$ as $T_{\texttt{shift}} \circ T_f$. This corresponds to the function $g$ we are looking for – $T_g$ operates in the same input and output space as $T_f$, so $g \in \mathcal{R}_\Sigma$; furthermore, this function satisfies for all $\boldsymbol{x} \in \mathcal{X}$ and $i \in [1, n-1]$: $g(\boldsymbol{x})_i = \texttt{shift}(f(\boldsymbol{x}))_i = f(\boldsymbol{x})_{i+1}$. $\qquad\square$

**Corollary 1.1.** Theorem 1 implies that when we parameterize a masked-diffusion model with a shifted transformer, it is as expressive as the standard non-shifted parameterization. More specifically, masked diffusion is defined in Equation (4), and $p_{\boldsymbol{\theta}}(x_i \mid \boldsymbol{x}^t)$ is typically implemented as a fully-bidirectional transformer model that outputs this probability at the $i$th position. When we set $\Sigma$ as our subword vocabulary, we get that the space of all possible transformer realizations of $p_{\boldsymbol{\theta}}(x_i \mid \boldsymbol{x}^t)$ are the $\Sigma$-realizable functions $\mathcal{R}_\Sigma$ (Definition 2). Theorem 1 shows that if we instead expect the output at the $(i-1)$th position, we do not lose any expressivity. Thus, transformer-based dual-objective language models are a generalization of standard masked-diffusion language models. Note that the left-shift closure in Theorem 1 works up to the first token – which is guaranteed to be the special  token in Definition 2 as well as in the actual implementation.

## G    VALIDATION LOSS CURVES

While we focused on actual downstream performance in the main experiments, we also show the validation loss below to demonstrate the training dynamics.

The validation curves in Figure 6 focus on an extremely data-constrained scenario with 128 data repetitions. There, it is crucial to avoid overfitting, which can be achieved by increasing the proportion of masked diffusion during training. Note that the noise of some of the curves is only due to our implementation of measuring the validation loss – the sample size can be too small when the proportion of the respective training objective is low.

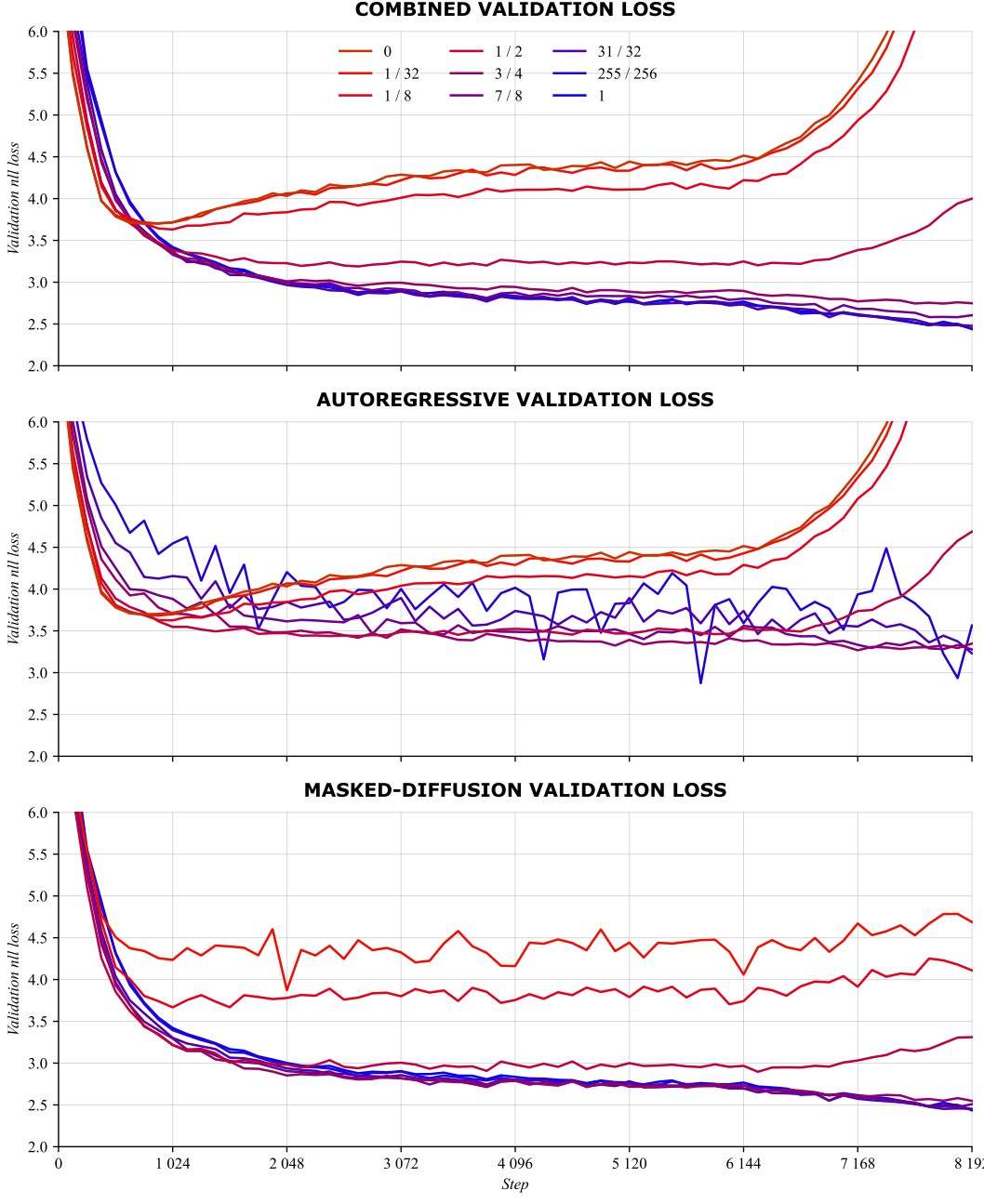

Figure 6: **Validation loss curves for 128 repetitions.** These plots clearly demonstrate how training runs with high $\alpha$ (in red) overfit. Low $\alpha$ values are in blue.

Contrary to the previous figure, Figure 7 shows validation curves for 4 data repetitions. Here, overfitting is not an issue; instead it is crucial to improve the learning speed by increasing the proportion of autoregressive language modeling.

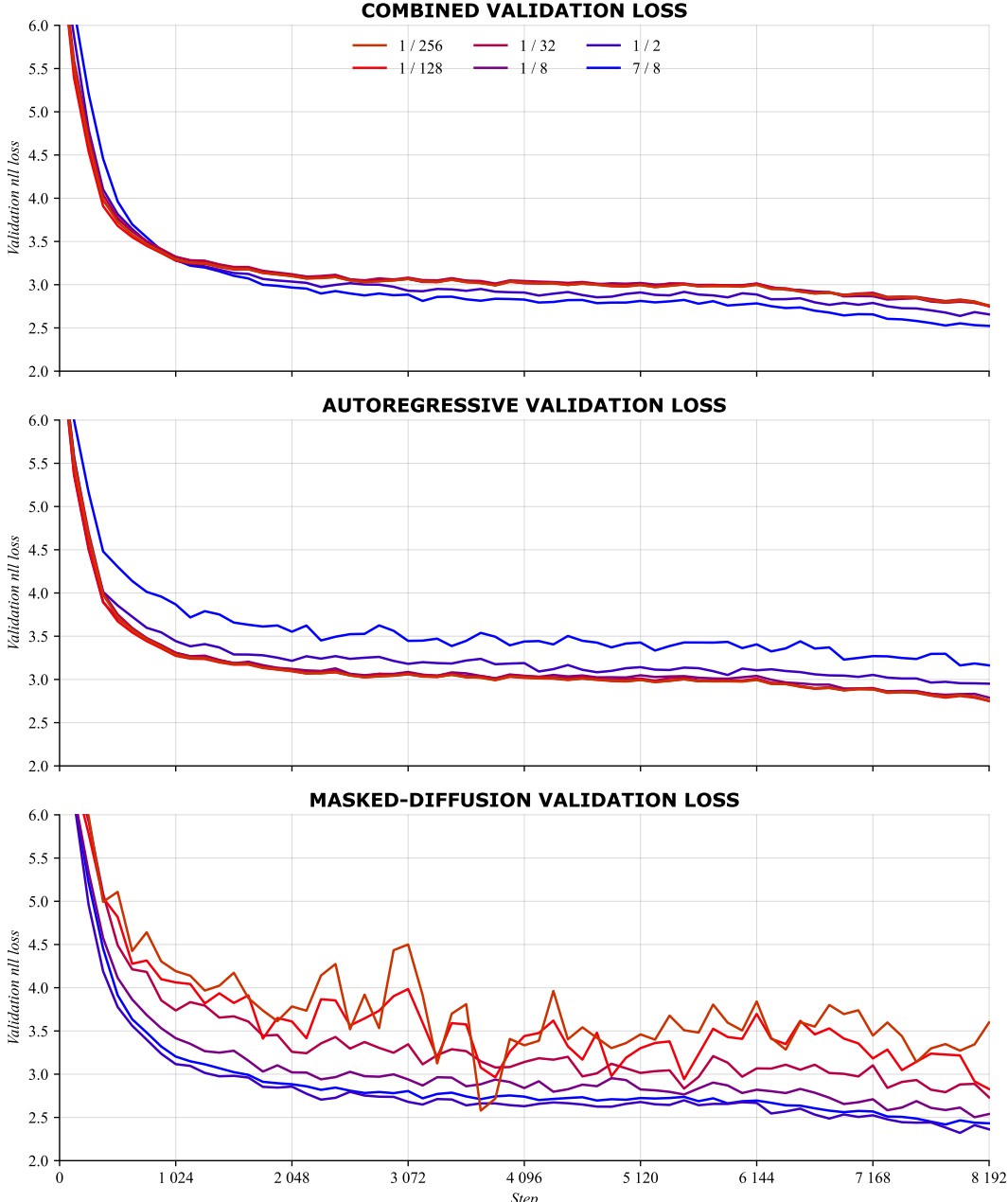

Figure 7: **Validation loss curves for 4 repetitions.** All losses monotonically decrease because overfitting is not a concern in this setting. High $\alpha$ values are plotted in red and low $\alpha$ values are shown in blue.

## H  EFFECTS OF NUMBER OF MASK TOKENS ON THE PLL

We first look at whether using a single number of mask tokens can lead to a good estimation of the PLL in general. For this, we evaluate five different models from 1 to 6 mask tokens and report the results in Tables 3 to 7.

Table 3: **PLL performance depending on the number of mask tokens.** We show the PLL performance on the 9 tasks of the model trained with an equal $\alpha$ weight of masked-diffusion and AR and 32 repetitions with different number of masks. Best results per task are boldfaced.

| Task | Number of masks | | | | | |
|---|---|---|---|---|---|---|
| | 1 | 2 | 3 | 4 | 5 | 6 |
| ARC Easy | 18.7 | 24.1 | 25.5 | **26.3** | 26.0 | **26.3** |
| ARC Challenge | **4.7** | 3.3 | 3.8 | 2.7 | 1.9 | 2.6 |
| BLiMP | **65.2** | 63.9 | 62.5 | 60.3 | 60.5 | 60.3 |
| Commonsense QA | 29.4 | 32.8 | 33.9 | **34.1** | **34.1** | **34.1** |
| HellaSwag | **29.8** | 27.0 | 26.7 | 27.1 | 26.7 | 26.4 |
| MMLU | 2.0 | **3.5** | 3.1 | 2.9 | 3.3 | 3.3 |
| OpenBook QA | 9.1 | 7.7 | 8.5 | **9.3** | 7.2 | 6.9 |
| PIQA | 33.1 | 34.3 | 35.1 | 35.4 | 35.6 | **36.8** |
| SIQA | 11.4 | 13.3 | 13.7 | 13.5 | **14.4** | **14.4** |
| **Average** | 22.6 | 23.3 | **23.6** | 23.5 | 23.3 | 23.4 |

Table 4: **PLL performance depending on the number of mask tokens.** We show the PLL performance on the 9 tasks of the model trained with a 1 masked-diffusion to 7 autoregressive ratio ($\alpha = 7/8$) and 32 repetitions with different number of masks. Best results per task are boldfaced.

| Task | Number of masks | | | | | |
|---|---|---|---|---|---|---|
| | 1 | 2 | 3 | 4 | 5 | 6 |
| ARC Easy | 18.2 | 25.6 | 27.1 | 28.2 | 26.9 | **27.5** |
| ARC Challenge | 1.9 | 3.2 | 2.4 | 2.6 | 3.6 | **4.7** |
| BLiMP | **61.2** | 60.0 | 58.3 | 56.9 | 57.0 | 57.3 |
| Commonsense QA | 24.2 | 29.1 | 29.0 | **29.4** | **29.4** | **29.4** |
| HellaSwag | 25.2 | 25.7 | 26.6 | 27.0 | 26.8 | 26.8 |
| MMLU | 1.9 | 3.4 | 4.0 | 3.9 | 4.0 | **4.2** |
| OpenBook QA | 9.9 | 10.1 | **12.3** | 10.9 | 10.1 | 9.6 |
| PIQA | 31.0 | 34.7 | **36.1** | 36.0 | 35.0 | 35.9 |
| SIQA | 11.7 | 11.8 | 14.2 | 13.7 | 14.1 | **14.3** |
| **Average** | 20.6 | 22.6 | **23.3** | 23.2 | 23.0 | **23.3** |

Table 5: **PLL performance depending on the number of mask tokens.** We show the PLL performance on the 9 tasks of the model trained with a 7 masked-diffusion to 1 autoregressive ratio ($\alpha = 1/8$) and 32 repetitions with different number of masks. Best results per task are boldfaced.

| Task | Number of masks | | | | | |
|---|---|---|---|---|---|---|
| | 1 | 2 | 3 | 4 | 5 | 6 |
| ARC Easy | 16.3 | 20.8 | 23.9 | 24.0 | **24.9** | **24.9** |
| ARC Challenge | **5.7** | 3.9 | 3.5 | 1.8 | 3.3 | 2.2 |
| BLiMP | **69.5** | 67.6 | 64.0 | 60.7 | 60.1 | 60.1 |
| Commonsense QA | 25.4 | 29.7 | 30.6 | 31.1 | 31.1 | **31.2** |
| HellaSwag | **25.5** | 22.8 | 21.0 | 21.2 | 20.5 | 19.8 |
| MMLU | 0.5 | 2.2 | 2.2 | 2.0 | **2.5** | 2.4 |
| OpenBook QA | 13.1 | 12.0 | **15.2** | 14.4 | 13.1 | 13.9 |
| PIQA | 29.6 | 30.3 | 30.8 | 30.1 | **31.2** | 31.0 |
| SIQA | 12.2 | 15.0 | **15.2** | 13.6 | 13.8 | 13.9 |
| **Average** | 22.0 | 22.7 | **22.9** | 22.1 | 22.3 | 22.2 |

Table 6: **PLL performance depending on the number of mask tokens.** We show the PLL performance on the 9 tasks of the model trained with an equal ratio of masked-diffusion and AR ($\alpha = 1/2$) and 16 repetitions with different number of masks. Best results per task are boldfaced.

| Task | Number of masks | | | | | |
|---|---|---|---|---|---|---|
| | 1 | 2 | 3 | 4 | 5 | 6 |
| ARC Easy | 16.8 | 23.7 | 25.8 | 25.8 | **26.1** | **26.1** |
| ARC Challenge | **7.2** | 4.4 | 4.4 | 4.8 | 3.2 | 4.5 |
| BLiMP | **65.3** | 64.8 | 63.1 | 60.7 | 60.6 | 60.4 |
| Commonsense QA | 29.7 | 33.8 | 35.1 | 35.1 | **35.2** | **35.2** |
| HellaSwag | **30.5** | 27.9 | 27.8 | 27.9 | 27.2 | 26.8 |
| MMLU | 1.3 | 2.4 | **2.9** | 2.5 | 2.7 | 2.5 |
| OpenBook QA | 12.3 | 12.0 | **13.1** | 11.2 | 11.7 | 11.7 |
| PIQA | 33.8 | 34.6 | 36.0 | 34.7 | 36.3 | **37.0** |
| SIQA | 14.3 | 13.9 | 15.9 | 15.3 | 15.9 | **16.1** |
| **Average** | 23.5 | 24.2 | **24.9** | 24.2 | 24.3 | 24.5 |

Table 7: **PLL performance depending on the number of mask tokens.** We show the PLL performance on the 9 tasks of the model trained with an equal ratio of masked-diffusion and AR ($\alpha = 1/2$) and 64 repetitions with different number of masks. Best results per task are boldfaced.

| Task | Number of masks | | | | | |
|---|---|---|---|---|---|---|
| | 1 | 2 | 3 | 4 | 5 | 6 |
| ARC Easy | 16.6 | 21.8 | 23.5 | 23.4 | 23.1 | 23.1 |
| ARC Challenge | 1.8 | 3.9 | 3.9 | 3.2 | **4.0** | 3.5 |
| BLiMP | **63.1** | 61.2 | 59.6 | 57.5 | 56.9 | 56.9 |
| Commonsense QA | 24.6 | 27.6 | 28.5 | **28.7** | **28.7** | **28.7** |
| HellaSwag | **26.8** | 25.2 | 24.2 | 24.7 | 24.3 | 24.1 |
| MMLU | 1.2 | 3.1 | 3.0 | 3.2 | **3.4** | 3.2 |
| OpenBook QA | 8.3 | 8.5 | **11.7** | 10.1 | 8.3 | 8.0 |
| PIQA | 31.0 | 31.7 | 32.1 | 33.7 | **34.3** | 34.1 |
| SIQA | **14.3** | 12.3 | **14.3** | 13.1 | 13.3 | 13.5 |
| **Average** | 20.8 | 21.7 | **22.3** | 22.0 | 21.8 | 21.7 |

We can see two clear trends from the results. The first is that the BLiMP and HellaSwag tasks are better evaluated with a single mask token, rather than multiple. This could be due to the simpler language found in these datasets. The second trend is that ARC-Easy, Commonsense QA, PIQA, and SIQA tend to do better with multi-token masking, this could be due to the more complex answers using more infrequent words that have a higher likelihood of being split into subwords. We therefore decide that using a combination of a single token mask for some tasks and a multiple tokens for others is the best solution. To find the optimal combination, we test all possible combinations. The results can be seen in Table 8.

Table 8: **PLL performance for combinations of one mask token and multi-mask token.** Best results per model are boldfaced.

| Repetitions – $\alpha$ | Mask combination | | | | |
| | $1-2$ | $1-3$ | $1-4$ | $1-5$ | $1-6$ |
|---|---|---|---|---|---|
| $32 - \frac{1}{2}$ | 24.1 | 24.5 | 24.6 | 24.7 | **24.8** |
| $32 - \frac{7}{8}$ | 22.8 | 23.6 | 23.7 | 23.5 | **23.8** |
| $32 - \frac{1}{8}$ | 23.5 | **24.3** | 24.0 | 24.1 | 24.2 |
| $16 - \frac{1}{2}$ | 24.9 | 25.7 | 25.4 | 25.7 | **25.8** |
| $64 - \frac{1}{2}$ | 22.3 | **23.0** | 22.9 | 22.9 | 22.8 |

Based on Table 8, we decide to evaluate PLL for all models with both a single mask token and six mask tokens. Then we take the max performance between the two for each task.

## I  TOTAL COMPUTE RESOURCES USED FOR TRAINING

The training of all 50 language models used in this paper was conducted on the LUMI supercomputer, each language model was trained on 128 AMD MI250X GPUs (which is equivalent to 256 logical devices) using roughly 1 500 GPU hours. In total, the resources required for conducting all training runs equals to approximately 75 000 GPU hours.

## J  PLL VERSUS MASKED DIFFUSION

Table 9 shows that the performance of the masked-diffusion model is in general lower than that of the combined (1 and 6 mask) PLL. In addition, the two PLL evaluations took about 2 hours to complete while the masked-diffusion evaluation takes 12 hours to complete on a MI250X AMD GPU.

Table 9: **Normalized PLL versus Masked-Diffusion evaluation.** The scores for each task are normalized so that 0% corresponds to the random baseline and 100% is the perfect score. The best result for each task is in boldfaced. We evaluate a model trained with equal AR and masked-diffusion ratio ($\alpha = \frac{1}{2}$) and 32 repetitions.

| Task | PLL | Masked-Diffusion |
|---|---|---|
| ARC-Easy | 26.3 | **27.1** |
| BLiMP | **65.2** | 56.5 |
| Commonsense QA | **34.1** | 32.7 |
| HellaSwag | **29.8** | 21.3 |
| PIQA | **36.8** | 32.0 |

## K  PREFIX-LM VERSUS AUTOREGRESSIVE-LM ON OPTIMAL MODELS.

Table 10 shows that evaluating with the prefix mask almost always outperforms using the causal mask when the models are optimally trained. This is true in both the regular and constrained data settings.

Table 10: **Normalized autoregressive and prefix performance of selected models.** The scores for each task are normalized so that 0% corresponds to the random baseline and 100% is the perfect score. The best result for each dataset size is in boldfaced. The results for BLiMP are the same, since there is no context and the prefix evaluation defaults to the autoregressive one. The AR ratios for the models are 12.5% for the 128 repetitions, 75% for the 32 repetitions, and 98.4% for the single repetition.

| Model | ARC-C | ARC-E | BLiMP | CSQA | HSwag | MMLU | OBQA | PIQA | SIQA | Average |
|---|---|---|---|---|---|---|---|---|---|---|
| 1 REPETITION | | | | | | | | | | |
| Autoregressive | 5.7 | 28.6 | **63.7** | 35.1 | 31.1 | 4.9 | **17.6** | 40.9 | 14.3 | 26.9 |
| Prefix | **6.5** | **31.0** | 63.7 | **40.0** | **31.2** | 4.5 | 16.5 | **42.1** | **15.2** | **27.9** |
| 32 REPETITIONS | | | | | | | | | | |
| Autoregressive | 3.3 | 28.0 | **57.9** | 31.1 | 26.4 | 3.6 | 14.4 | 36.1 | 14.64 | 23.9 |
| Prefix | **6.3** | **28.9** | 57.9 | **33.1** | **27.1** | **4.3** | **15.2** | **36.7** | **15.4** | **25.0** |
| 128 REPETITIONS | | | | | | | | | | |
| Autoregressive | **1.7** | 23.6 | **56.1** | 24.8 | 14.2 | 1.6 | 8.5 | 28.1 | 13.3 | 19.1 |
| Prefix | 1.3 | **24.1** | 56.1 | **28.5** | 12.4 | **2.3** | **10.9** | **30.9** | **15.2** | **20.5** |

## L   DETAILED RESULTS OF DIFFUSION-MASKED EVALUATION

Table 11: **The normalized PLL performance of selected models.** We show the results on all nine evaluated tasks for three repetition values; each repetition group contains the results of the best-performing $\alpha$ and of the autoregressive-only model. The scores for each task are normalized so that 0% corresponds to random baseline and 100% is the perfect score. The best result for each dataset size is boldfaced.

| Model configuration | ARC-C | ARC-E | BLiMP | CSQA | HSwag | MMLU | OBQA | PIQA | SIQA | Average |
|---|---|---|---|---|---|---|---|---|---|---|
| 32 REPETITIONS | | | | | | | | | | |
| Dual ($\alpha = 3/4$) | **6.0** | **28.3** | 62.7 | **33.4** | **27.8** | **4.3** | **12.3** | **37.4** | **15.4** | **25.3** |
| Masked-Diffusion ($\alpha = 0$) | -0.1 | 22.3 | **64.8** | 29.0 | 24.1 | 1.6 | 9.1 | 27.2 | 14.4 | 21.4 |
| 128 REPETITIONS | | | | | | | | | | |
| Dual ($\alpha = 1/8$) | 2.8 | **23.3** | **63.5** | **30.5** | **25.0** | 2.1 | **12.8** | **31.8** | **15.2** | **23.0** |
| Masked-Diffusion ($\alpha = 0$) | **3.3** | 19.2 | 63.3 | 29.2 | 22.1 | **2.6** | 9.3 | 28.3 | 12.0 | 21.0 |

Table 11 shows similar trends to those found in Table 2. The notable exception being for BLiMP where the performances are similar between both models. Unlike the autoregressive models, the performance of the purely masked-diffusion models is similar to each other. This is partially due to the model not overfitting, but also to it not being sample efficient. On the other hand we see that for the dual-objective models, the performance significantly increases as we increase the training data set size.

