# OpenReview forum: "Dual-objective Language Models: Training Efficiency Without Overfitting"
_ICLR.cc/2026/Conference — ICLR 2026 Poster_

### Official Review · Reviewer_Z29n · 2025-10-17

**Soundness:** 2
**Presentation:** 3
**Contribution:** 3
**Rating:** 6
**Confidence:** 3

**Summary:**

This paper proposes Dual Language Models (Dual-LMs), which jointly train a single transformer with both autoregressive (AR) and masked-diffusion (MD) objectives. The key motivation is to balance sample efficiency (of AR training) and overfitting resilience (of MD training) without changing the model architecture. The authors conduct a large-scale empirical study on 50 language models to support the method's effectiveness.

**Strengths:**

1. Clear motivation and timely problem.
The work addresses a relevant issue as large-scale language model training increasingly encounters limited high-quality data and potential overfitting under repeated exposure.
2. Practical recommendations
The derived empirical guidelines (e.g., small MD component under regular regimes, stronger MD ratio under high repetition) are actionable for practitioners designing large-scale LLM training recipes.

**Weaknesses:**

1. Limited contribution in standard settings.
When trained with only 1× data repetition, Dual-LMs provide no significant improvement over pure autoregressive baselines. This weakens the overall claim of universal benefit; the method is primarily helpful under extreme data-reuse scenarios
2. Practical rarity of extreme repetition.
The key advantage appears only under 128× repetition, which is rarely used in modern large-scale LLM training. In realistic compute-optimal regimes (1–8× repetition), the dual-objective gains are negligible.
3. Learning rate confound.
For very high repetition, a smaller learning rate is typically required to maintain stability. Since the experiments maintain a fixed LR schedule, it is unclear whether the observed benefits of Dual-LMs come from the objective mixing itself or from suboptimal tuning in baselines.
4. Method scope and positioning.
Overall, the approach seems more suitable as a regularization mechanism for repeated training rather than a fundamentally new paradigm for LLM optimization. The contribution is empirical rather than conceptual, and its practical importance outside data-scarce settings is limited.

**Questions:**

1. The paper’s results suggest that the dual-objective helps mainly under extreme data repetition. What prevents it from providing gains in standard (1–8×) training regimes? Is it fundamentally tied to overfitting mitigation rather than general sample efficiency improvement?

---

> ### Author Response · Authors · 2025-11-24
>
> We would like to thank the reviewer for the thorough feedback. We also appreciate that the reviewer acknowledges the impact of this work and its practicality.
>
> _____
>
> **Weakness 1: When trained with only 1× data repetition, Dual-LMs provide no significant improvement over pure autoregressive baselines.**
>
> We respectfully disagree with this assessment. Our results demonstrate that dual-objective training provides consistent benefits even with a single data repetition. This is however an important feedback for us, we will consider more clearly highlighting the improved masked-diffusion performance.
>
> Examining Figure 4(a,b) at the leftmost columns ($1\times$ repetition), we observe that dual models achieve:
> - superior masked-diffusion performance compared to pure MD models, without sacrificing any autoregressive performance;
> - equivalent autoregressive performance to pure AR models, while additionally gaining strong bidirectional capabilities.
>
> This represents a significant practical advantage: practitioners obtain a single model excelling at both unidirectional and bidirectional tasks without any computational overhead. We will revise our presentation to emphasize that the value proposition extends beyond AR metrics alone – the ability to perform both AR and MD tasks effectively with a single model is a key contribution.
>
> _____
>
> **Weakness 2: Practical rarity of extreme repetition.**
>
> We acknowledge this concern and agree that $128\times$ repetition represents an extreme scenario for English datasets at this moment. However, we emphasize two critical points:
> - First, high repetition is already a reality for the vast majority of world languages. Training large models necessitates enough data [1], which is currently only achievable by repeating the data [2].
> - Second, data constraints are becoming relevant even for high-resource languages. The well-documented scaling laws show that model improvements follow a power law with respect to data -- each incremental performance gain requires exponentially more tokens. As the field approaches the limits of available high-quality text (the "data wall"), methods robust to repetition become increasingly critical. Our work provides a principled solution before this limitation becomes acute. We reflect on this point in the newly added Appendix K (Extended related work).
>
> _____
>
> **Weakness 3: Learning rate confound. For very high repetition, a smaller learning rate is typically required to maintain stability.**
>
> We appreciate the reviewer raising this methodological question. However, we believe there may be a misunderstanding about the nature of the challenge with high repetition. The issue is not training instability (which might benefit from reduced learning rates) but rather overfitting, where the model memorizes training data. We have never encountered any training instability during the experimental training runs.
>
> Our loss curves in Appendix J clearly demonstrate that:
> - models with high AR ratios show classic overfitting patterns;
> - reducing learning rates would potentially exacerbate overfitting by allowing the model to more precisely memorize training examples;
> - the dual objective provides regularization that maintains validation performance.
>
> We use consistent learning rate schedules across all experiments to ensure fair comparison. Importantly, we mainly compare models trained at the same data repetition, not across repetitions.
>
> _____
>
> **Weakness 4: Method scope and positioning.**
>
> We believe our contribution is both practically significant and conceptually novel:
> - Practical significance: we provide the first systematic study showing that dual-objective training is optimal across all data regimes, with concrete recommendations for practitioners.
> - Conceptual contribution: we demonstrate that AR and MD objectives are not mutually exclusive but complementary, challenging the current paradigm of choosing one or the other.
> - Scalability: our method requires no architectural changes, making it immediately applicable to existing infrastructure.
>
> We acknowledge that our contribution is primarily empirical, but given the substantial performance improvements of masked diffusion and practical applicability, we believe this represents an important advance for the field.
>
> _____
>
> **Question 1: What prevents it from providing gains in standard (1–8×) training regimes?**
> As detailed in our response to Weakness 1, dual-objective training does provide gains in standard regimes, but these gains manifest primarily in:
> - acquiring bidirectional capabilities without sacrificing AR performance;
> - substantially improved masked-diffusion capabilities;
> - improved prefix language modeling (Section 5.4);
> - greater flexibility in downstream applications.
>
> _____
>
> [1] [Training compute-optimal large language models](https://dl.acm.org/doi/10.5555/3600270.3602446)
>
> [2] [Scaling data-constrained language models](https://dl.acm.org/doi/10.5555/3666122.3668313)

---

### Official Review · Reviewer_apc1 · 2025-10-29

**Soundness:** 3
**Presentation:** 3
**Contribution:** 2
**Rating:** 4
**Confidence:** 3

**Summary:**

This paper studies the trade-off between sample efficiency and overfitting resilience in language modeling by combining autoregressive and masked-diffusion objectives in a single transformer architecture, without architectural modifications. The authors conduct an extensive empirical study, training 50 models across varying data repetition regimes, to uncover the optimal balance between the two objectives. They show that the dual-objective approach consistently outperforms single-objective baselines in both autoregressive and masked-diffusion downstream tasks, providing practical guidance on loss balancing and implications for prefix language modeling.

**Strengths:**

- The authors train and evaluate 50 language model configurations under a wide range of data repetition regimes, allowing for robust conclusions on objective balancing.
- The dual-objective method is introduced without architectural changes, maximizing practical usability.
- The paper distills its complex results into two practical, easy-to-understand guidelines for training models in regular and data-constrained regimes.

**Weaknesses:**

- The core idea of combining AR and masked objectives in a single decoder-only model is not entirely new and builds heavily on previous work, particularly GPT-BERT, which the authors acknowledge. The primary contribution is more of an extensive empirical investigation rather than a fundamentally new training paradigm.
- While Section 6 discusses prior methods such as GPT-BERT and AntLM, the paper does not include empirical comparisons with these or other recent approaches (e.g. Block Diffusion [1], AR-Diffusion [2]) that address the same AR–diffusion trade-off. Including recent works as baselines under the same training setup would provide a clearer and fairer assessment of the proposed method’s advantages.
- The paper assumes that the findings generalize to larger models, but this remains speculative, as all experiments are limited to the 470M scale. Given the unpredictability of scaling behavior, it is unclear whether the observed trends would hold for models beyond 1B parameters.
- While Equations (2) and (4) define the individual objectives, the paper offers little theoretical explanation of how their joint optimization works. The rationale for specific weighting choices is empirical, and potential gradient interference or complementarity between the two losses is not analyzed.

[1] Arriola, Marianne et al. “Block Diffusion: Interpolating Between Autoregressive and Diffusion Language Models.” 2025
[2] Wu, Tong et al. “AR-Diffusion: Auto-Regressive Diffusion Model for Text Generation.” 2023

**Questions:**

- The paper builds on established ideas such as GPT-BERT and CM3, with the main contribution appearing to be the systematic analysis of data repetition effects. Could the authors clarify the specific technical or algorithmic innovations introduced by the proposed dual-objective framework beyond these prior methods?
- Would the authors consider adding or discussing comparisons with related works (e.g. Block Diffusion and AR-Diffusion), which also address the AR–diffusion trade-off?
- Can the authors provide evidence—empirical or analytical—that the optimal AR–MD ratios and the 16-repetition guideline (Remark 2) remain valid for larger models (≥ 1B parameters)?
- Can the authors provide a more rigorous theoretical basis or empirical ablation for their objective weighting strategy?

---

> ### Author Response · Authors · 2025-11-24
>
> We thank the reviewer for the thorough and constructive feedback. We appreciate the recognition of our work's practicality and of the robust evaluation. We used the feedback to improve the paper in the newly revised version. Below, we address each concern raised and provide additional clarification where needed.
>
> _____
>
> **Weakness 1: The core idea [...] builds heavily on previous work.**
>
> We acknowledge building on GPT-BERT's foundation throughout the paper, but our contributions go significantly beyond their preliminary work. Specifically:
> 1) we extend the approach to masked-diffusion models rather than masked language modeling;
> 2) we give a formal proof for the parameterization with next-token prediction (Appendix I);
> 3) we provide a systematic study across 50 model configurations and demonstrate that dual objectives are optimal across all evaluated settings, not just toy cases and small finetuned models as in GPT-BERT;
> 4) we focus on the problem of overfitting and provide a systematically evaluated solution;
> 5) we provide concrete training guidelines (Remarks 1, 2) relevant for modern language models.
>
> _____
>
> **Weakness 2: Empirical comparisons with Block Diffusion and AR-Diffusion.**
>
> We would like to thank the reviewer for highlighting these works that are indeed relevant. We have added them to our related work (Appendix K) in the revised paper. However, direct comparison would be methodologically inappropriate as these methods solve orthogonal problems. Block Diffusion and AR-Diffusion slightly modify diffusion models to improve their generation speed, while we propose a dual training objective that generalizes over autoregressive and masked-diffusion language modeling. Our dual-objective models could actually benefit from Block Diffusion's faster decoding -- the improvements are complementary, not competitive.
>
> _____
>
> **Weakness 3: The paper assumes that the findings generalize to larger models.**
>
> We deliberately chose the 470M scale to enable comprehensive evaluation across 50 configurations—scaling to 1B+ parameters would require a prohibitive amount of compute. However, our findings should generalize for two reasons: (1) The optimal ratios correlate with overfitting behavior, which the previous work [1] show is scale-invariant for autoregressive models, (2) The relative parametric burden of the dual objective decreases with model size, suggesting our benefits would increase at scale. But ultimately, this is a valid objection, we tried to already transparently address this concern in the paragraph titled “Generalization to larger language models” (line 370).
>
> _____
>
> **Weakness 4: Theoretical explanation of how their joint optimization works.**
>
> We apologize for the unclear description. The key insight is that Equations (2) and (4) are not separate objectives but two different *factorizations* of the same maximum likelihood objective (Equation 1). By mixing sequences with different masking patterns in each batch, we effectively optimize the same underlying MLE objective through both factorizations simultaneously.
>
> The weighting (2:1 for AR:MD) simply accounts for the mathematical fact that E[1/t] = 2 in the masked-diffusion ELBO, ensuring balanced gradient contributions. We have clarified this in the revised manuscript. Furthermore, we have now added a formal proof in Appendix I that the shifted (next-token) parameterization is as strong as the standard parameterization of masked-diffusion models.
>
> _____
>
> **Question 1: Could the authors clarify the specific technical or algorithmic innovations introduced?**
>
> We hope this is answered by our response to Weakness 1. Compared to CM3, our dual language models generalize over both autoregressive and masked-diffusion language models, so they can be used in the same way without any changes. Furthermore, our training objective does not require a specific structured input.
>
> _____
>
> **Question 2: Would the authors consider adding or discussing comparisons with related works (e.g. Block Diffusion and AR-Diffusion)?**
>
> As explained in response to Weakness 2, these methods are complementary to ours. We discuss these methods in the newly introduced Appendix K (to be added to the main text).
>
> _____
>
> **Question 3: Evidence for scalability.**
>
> Please see our response to Weakness 3.
>
> _____
>
> **Question 4: Can the authors provide a more rigorous theoretical basis or empirical ablation for their objective weighting strategy?**
>
> The necessity of the 1/t loss weighting is explained in detail by previous work on masked diffusion [2]; this is not our contribution, but established theory.
>
> _____
>
> [1] [Scaling data-constrained language models](https://dl.acm.org/doi/10.5555/3666122.3668313)
>
> [2]: [Your Absorbing Discrete Diffusion Secretly Models the Conditional Distributions of Clean Data](https://openreview.net/forum?id=sMyXP8Tanm)

---

### Official Review · Reviewer_59vc · 2025-10-29

**Soundness:** 4
**Presentation:** 4
**Contribution:** 4
**Rating:** 8
**Confidence:** 5

**Summary:**

This work studies Dual LM, which combines the auto aggressive next token prediction and masked language modeling into a single model. To be specific, authors are investigating the potential to train one single LLM with two different training objectives. And the finding is we can achieve a better trade off between compute efficiency and data efficiency by adjusting the ratio of two training objectives.

**Strengths:**

1) Very clear presentation. The lessons and insights are very presented so that readers can understand the problem well.
2) Studying the data efficiency is a very important problem, for both large and small models, especially considering we are overtraining models for better quality and cheaper serving cost.
3) The sweep on training objective ratio is informative. This can help to design better training schedule targeting on a specific data/compute efficiency.

**Weaknesses:**

1) The definition of sample-efficiency is confusing. I think it would be better to rephrase it as "compute-efficiency". The sample efficiency makes people confused whether the sample here means "unique token efficiency".
2) Minor: I understand the research is usually for long term purpose, but to be very honest, the conclusion we achieved in this work is we only need dual LM for very aggressive repeats. This is anyway informative but not something we have to use now.

**Questions:**

NA

---

> ### Author Response · Authors · 2025-11-24
>
> We thank the reviewer for their constructive feedback and we are happy for the appreciation of the impact of this work. We have improved the paper by  incorporating the suggested clarifications in the revised version of the paper.
>
> _____
>
> **Weakness 1: Definition of sample-efficiency.**
>
> We appreciate this clarification, it is a great point! We thought about the proper term here and “training efficiency” is possibly the most exact term -- definitely better than "sample efficiency".  We have updated this terminology throughout the revised manuscript (see the changes in blue). While "compute efficiency" could also be appropriate, we believe "training efficiency" better highlights that the compute efficiency is limited to training.
>
> _____
>
> **Weakness 2: We only need dual LM for very aggressive repeats**
>
> It is useful for us that the reviewer highlights this point, as it indicates that we should clarify our contributions more prominently. Our results demonstrate that the dual objective consistently outperforms single-objective training across all evaluated settings, not only under aggressive repetition. Specifically, even with single repetition (leftmost column in Figure 4a,b), dual training achieves superior masked-diffusion performance (compared to a directly trained masked-diffusion model) without losing any autoregressive performance. Thus, our method enables training a single model that excels at both AR and MD tasks without requiring separate models, reducing training costs and deployment complexity. We tried to highlight this aspect in Remark 1.
>
> While we acknowledge the increasing importance for data-constrained scenarios (which are quickly becoming more important, see the last paragraph in the newly added Appendix K), the immediate practical benefit of obtaining dual capabilities without performance degradation makes this approach valuable today.

---

> > ### Comment · Reviewer_59vc · 2025-11-27
> >
> > Thank you.
> > I decide to keep my score.

---

### Official Review · Reviewer_cMFP · 2025-11-06

**Soundness:** 2
**Presentation:** 3
**Contribution:** 3
**Rating:** 4
**Confidence:** 2

**Summary:**

This paper aims to introduce a unified training objective for language modeling that jointly optimizes autoregressive (AR) and masked-diffusion losses within a single transformer, while requiring no architectural modification.
By co-training a single transformer model on both losses, DLM leverages AR’s rapid convergence while regularizing with the diffusion objective to prevent overfitting. Extensive experiments across diverse benchmarks demonstrate that this hybrid objective consistently enhances performance over single-objective baselines.

**Strengths:**

This paper is well written and easy to follow. The narrative is coherent, and the topic is both timely and important for the current stage of large language model development.

The proposed DLM objective is conceptually simple and practically appealing. It requires no architectural changes and includes GPU-efficient adaptations, making the method easy to implement and broadly applicable. This design choice enhances the practical relevance of the work.

The empirical evaluation appears systematic and comprehensive. The experiments are carefully structured, covering a wide range of data regimes and objective ratios, which strengthens the credibility of the results and the generality of the conclusions.

Although the work is primarily empirical, it offers new insights that meaningfully advance understanding in the community.

**Weaknesses:**

### The link to diffusion language model is a bit weak

It is unclear to me whether the “diffusion mode” in the proposed objective truly corresponds to a standard masked diffusion language model, as to my knowledge, no published work implements diffusion language model using a decoder.
Conceptually, the proposed objective appears closer to introducing stochastic regularization into standard autoregressive training—adding noise to the input sequence to improve generalization and reduce overfitting.
Moreover, it remains ambiguous whether the resulting model supports parallel decoding, a defining property of diffusion-based LMs.
While the paper claims that the model “can generalize to prefix LM,” it does not explicitly demonstrate diffusion-style generation. A clearer explanation of how the diffusion time-conditioning interacts with the next-token prediction objective would strengthen the technical grounding.


### Lack of analytical justificaiton.

The paper’s conclusions are derived almost entirely from empirical evidence. While the experiments are thorough and convincing, additional theoretical or analytical insights would improve the work’s depth. For example, identifying sufficient conditions (some toy cases) under which the dual-objective training is guaranteed to outperform either individual objective would help explain why the method works beyond observation.


### Incomplete discussion of training dynamics.
The paper focuses primarily on zero-shot, multiple-choice benchmarks but provides limited discussion of training behavior. Analyses of the loss curves (for both AR and diffusion modes), convergence patterns, and stability would give a more complete understanding of how the dual objective influences optimization and generative quality.

**Questions:**

* Could you elaborate more on how closely does the proposed “diffusion mode” correspond to a standard masked-diffusion language model? For instance, how is the diffusion time variable integrated into the next-token prediction objective, is it sampled per token or globally for the sequence, and how does this choice affect model behavior? How does the new objective support parallel decoding or multi-token prediction?

* How stable is the training process under the dual objective? I am wondering what would the loss curves look like and whether there are any significant tradeoffs between the two modes during training?

* How does this work relate to a recent work [1] that proposed a modified decoder architecture to do any-order next-token prediction?

[1] Any-Order GPT as Masked Diffusion Model: Decoupling Formulation and Architecture, arxiv preprint 2506.19935

---

> ### Author Response · Authors · 2025-11-24
>
> We appreciate that the reviewer has carefully gone through our paper and raised valid concerns. We have revised and improved the paper based on the feedback (changes are highlighted in blue, the new sections are in Appendices I, J and K for now). We also thank the reviewer for acknowledging the practical impact and robust evaluation of our method.
>
> _____
>
> **Weakness 1: The link to diffusion language model is a bit weak**
>
> This is important feedback. The reviewer is absolutely right that we should have made this connection more theoretically grounded. To address this concern, we have now added Appendix I that formally proves that our parameterization (bidirectional transformer with shifted outputs) is as expressive as the standard parameterization of masked-diffusion models. Furthermore, note that our "diffusion mode" implements the exact masked-diffusion objective from prior work, as shown in Equation (4).
>
> Regarding terminology: The terms “encoder” and “decoder” can be confusing in this context. We have tried to avoid using them in the paper, instead we talk about transformer (blocks) that are either masked causally, or use the full bidirectional attention.
>
> For parallel decoding: Yes, the model supports standard masked-diffusion parallel decoding during inference. The great benefit of dual language models is that they can be used in the same way as any standard autoregressive or masked-diffusion models.
>
> _____
>
> **Weakness 2: Lack of analytical justification.**
>
> We believe that our idea is clearly motivated and theoretically sound (Appendix I). We completely agree with the reviewer about the value of proper analytical justification. However, this is currently impossible for modern LLMs, their theoretical analysis remains an open challenge across the field, even for much simpler training objectives than ours.
>
> _____
>
> **Weakness 3: Incomplete discussion of training dynamics.**
>
> This is a valid concern. We have added Appendix J with detailed validation loss curves showing the training dynamics under different objective ratios. The curves reveal that:
> - with high autoregressive ratios, validation loss initially decreases rapidly but then diverges (overfitting);
> - with high masked-diffusion ratios, convergence is stable but slow;
> - the dual objective maintains stable convergence while achieving lower final validation loss;
> - the training process is notably stable - we observe no optimization difficulties across all 50 trained models. Both training objectives decrease monotonically.
>
> _____
>
> **Question 1: Could you elaborate more on how closely does the proposed “diffusion mode” correspond to a standard masked-diffusion language model?**
>
> [Merged with response to Weakness 1 above for coherence]
>
> _____
>
> **Question 2: How stable is the training process under the dual objective?**
>
> Hopefully answered by the response to Weakness 3 and the new Appendix J. We have not found any issues with instability.
>
> _____
>
> **Question 3: How does this work relate to a recent work that proposed a modified decoder architecture to do any-order next-token prediction?**
>
> This is a valuable reference, we would like to thank the reviewer for pointing us to this very recent work. We have added discussion in Appendix K (to be moved to main text for camera-ready). Xue et al. (2025) modifies masked-diffusion by using causally-masked transformers, decoupling architectural differences from training-objective differences. Their finding that masked-diffusion alone underperforms on modeling language actually supports our approach - we show that combining objectives addresses precisely this limitation while maintaining architectural simplicity.

---

### Meta-Review · Area_Chair_ssAb · 2026-01-06

**Summary:**

This paper proposes training language models with both the masked token prediction as well as the next token prediction objectives with a shared architecture. It performs systematic empirical evaluations of 50 LMs and demonstrate that this dual objective is beneficial in certain regimes (eg high data repetition). It receives mixed reviews: on the one side most reviewers acknowledge the value of the proposition and appreciate the comprehensive evaluations, on the other hand there are also questions regarding novelty, practicality and positioning.

The AC believes that there is great merit in the proposed idea, especially given the current state of LLM scaling. The authors also did a fairly good job responding to the weaknesses, and overall the AC believes that it's a valuable contribution to the community.

**Reviewer Concerns:**

All issues related to positioning/presentation seem to be addressed or can be fixed in the updated version.

There are inherent limitations, such as the setting itself is not entirely novel and that the findings are not immediately practical. But the AC believes that's just the nature of this work and should not be reasons to reject it.

**Reviewer Scores:**

I'd expect rev cMFP and Z29n willing to increase their ratings while the other two had stated that they would maintain their ratings.

---

### Decision · Program_Chairs · 2026-01-26

Accept (Poster)